

# Comparison of published palaeoclimate records suitable for reconstructing annual to sub-decadal hydroclimatic variability in eastern Australia: implications for water resource management and planning

Anna L. Flack[1], Anthony S. Kiem[1], Tessa R. Vance[2], Carly R. Tozer[2,3], and Jason L. Roberts[4]

[1]Centre for Water, Climate and Land (CWCL), Faculty of Science, University of Newcastle, Callaghan, NSW 2308, Australia
[2]Institute for Marine and Antarctic Studies (previously Antarctic Climate and Ecosystems Cooperative Research Centre (ACE CRC)), University of Tasmania, Hobart, Tasmania 7004, Australia
[3]CSIRO Oceans & Atmosphere, Hobart, Tasmania 7004, Australia
[4]Australian Antarctic Division, Kingston, Tasmania 7050, Australia

*Correspondence to*: Anthony S. Kiem (Anthony.Kiem@newcastle.edu.au)

**Abstract.** Knowledge of past, current and future hydroclimatic risk is of great importance. However, like many other countries, Australia's observed hydroclimate records are at best only ~120 years long (i.e. from ~1900 to present) but are

typically less than ~50 years long. Therefore, recent research has focused on developing longer hydroclimate records based on palaeoclimate information from a variety of different sources. Here we review and compare the insights emerging from 11 published palaeoclimate records that are relevant to annual to sub-decadal hydroclimatic variability in eastern Australia over the last ~1000 years. The sources of palaeoclimate information include ice cores, tree rings, cave deposits and lake sediment deposits. The published palaeoclimate information was then analysed to determine when (and where) there was

agreement (or uncertainty) about the timing of wet and dry epochs in the pre-instrumental period (1000-1899). The occurrence, frequency, duration and spatial extent of pre-instrumental wet and dry epochs was then compared to wet and dry epochs since 1900. The results show that instrumental records (~1900-present) underestimate (or at least misrepresent) the full range of rainfall variability that has occurred, and is possible, in eastern Australia. Even more disturbing is the suggestion, based on insights from the published palaeoclimate data analysed, that 71% of the pre-instrumental period

appears to have no equivalent in the instrumental period. This implies that the majority of the past 1000 years was unlike anything encountered in the period that informs water infrastructure, planning and policy in Australia. A case study, using a typical water storage reservoir in eastern Australia, demonstrates that current water resource infrastructure and management strategies would not cope under the range of pre-instrumental conditions that this study suggests has occurred. When coupled with projected impacts of climate change and growing demands, these results highlight some major challenges for water

resource management and infrastructure. Though our case study location is eastern Australia, these challenges, and the limitations associated with current methods that depend on instrumental records that are too short to realistically characterise interannual to multidecadal variability, also apply globally.



## 1 Introduction

Knowledge of drought and flood history is of great importance and has many implications for current and future water
resource management, especially for the densely populated east coast of Australia. Like many other countries, most
instrumental rainfall records in Australia only exist for ~120 years at best, and streamflow records longer than ~50 years are
rare. During this short time, Australia has experienced some serious droughts and floods. For example, the Millennium
drought, which lasted from ~1997 to 2010, and during which urban and rural water resources in eastern Australia were put
under significant stress (Verdon-Kidd and Kiem, 2009; Kiem et al., 2016), and the 2010-2011 Queensland floods where the
Wivenhoe Dam reached capacity, overflowed and destroyed dozens of properties (http://www.floodcommission.qld.gov.au/)
(Van den Honert and McAneney, 2011; McMahon and Kiem, 2018). To enable design and implementation of robust
adaptation and management plans it is important to properly understand the chance of similar or worse droughts and floods
happening again. To date, in Australia and elsewhere, methods to quantify drought/flood frequency and risk have relied
primarily on instrumental records. However, recent research demonstrates that the instrumental period is not long enough to
get a true indication of the variability possible or the risk of extreme hydrological events (e.g. prolonged drought or flood
dominated epochs) in, for example:

- Australia (e.g. Gallant and Gergis, 2011; Ho et al., 2014; Allen et al., 2015a, 2015b; Ho et al., 2015a, 2015b; Vance et al., 2015; Tozer et al., 2016, 2018; Dixon et al., 2017, 2019; Kiem et al., 2020);
- Asia (e.g. Davi et al., 2013; Feng et al., 2013; Pederson et al., 2014; Nguyen and Galelli, 2018; Rao, 2018; Wang et al., 2019);
- the United States of America (USA) (e.g. Cook et al., 2004; Littell et al., 2016; Martin et al., 2019; Robeson et al., 2020);
- Europe (e.g. Cook et al., 2015; Perşoiu et al., 2017; Hanel et al., 2018);
- South America (e.g. Lara et al., 2008; Urrutia et al., 2011; Barria et al., 2018; Fernández et al., 2018).


Climate variability in the pre-instrumental past can be inferred from palaeoclimate proxies. However, in the Southern
Hemisphere and eastern Australia in particular, there are limited palaeoclimate records available, especially in comparison to
the Northern Hemisphere (Ho et al., 2014; Neukom et al., 2014). Therefore, it is important to determine how the few
palaeoclimate records that exist, and are relevant in terms of variables reconstructed and temporal/spatial resolution, can be
best utilized to infer pre-instrumental hydroclimatic histories for locations within eastern Australia. Pre-instrumental
hydroclimate reconstructions for Australia have been attempted using palaeoclimate proxies including tree rings, ice cores,
corals and lake sediments (e.g. Lough, 2011; Vance et al., 2013; Barr et al., 2014; Allen et al., 2015a; Palmer et al., 2015;
Allen et al., 2017). Due to the lack of in-situ (local) proxies in Australia, many of these reconstructions rely on using remote
proxies (i.e. proxies outside of Australia or the catchment of interest). Remote proxies utilize a climate teleconnection
between the location of the proxy record and the location of the target climate variable (e.g. a water catchment).





Currently there is no comprehensive comparison of where and when the few palaeoclimate records that exist for Australia agree or disagree with respect to pre-instrumental hydroclimatic conditions for eastern Australia. This study reviews and compares the insights emerging from 11 published palaeoclimate records that are relevant to annual to sub-decadal hydroclimatic variability in eastern Australia over the last ~1000 years. Based on where and when the pre-instrumental records agree, the occurrence, frequency, duration and spatial extent of pre-instrumental (1000-1899) wet and dry epochs are inferred and this is then compared with the same characteristics derived from the instrumental period (1900-1999). A case study, using a typical water storage reservoir in eastern Australia, then demonstrates some implications of the insights emerging from the pre-instrumental data and discusses the challenges posed for water resources management and planning in a variable and changing climate. Though our case study location is eastern Australia, these challenges, and the limitations associated with current methods that depend on instrumental records that are too short to realistically characterise interannual to multidecadal variability, also apply globally.

## 2 Selecting published palaeoclimate records that are relevant to annual to sub-decadal hydroclimatic variability in eastern Australia

Palaeoclimate information can be sourced from local or remote proxies. Local proxies provide climate records directly from or applicable to a target location, for example cave deposits within a study region/catchment or local tree ring records (e.g. McDonald et al., 2007; Heinrich et al., 2009; Allen et al., 2015b; Ho et al., 2015b). Remote proxies are located outside the region/catchment of interest, and maybe outside of Australia (e.g. Vance et al., 2015; Palmer et al., 2015).

Table 1 shows the 11 palaeoclimate records that were selected and analysed in this study. The palaeoclimate records used were selected from palaeoclimate information available (i.e. published) at the time of study, based on the following criteria:

- *Spatial extent*. Records must be sourced from locations within eastern Australia (if a local record) or be relevant to eastern Australia (if a remote record). Figure 1 shows the spatial extent (or relevant location) for the palaeoclimate records used in this study. Figure 1 also indicates the type of proxy used and the type of information provided. The information may be in the form of a rainfall/streamflow reconstruction (or inference) or an indication of dry or wet epochs over time.

- *Temporal resolution*. Records with annual resolution were preferred (9 out of 11 records), however, two records with ~4-5 year temporal resolution were also used as this is consistent with this study's focus on sub-decadal hydroclimatic variability.

- *Time period covered*. Records used need to provide data for some or all of the 1000-1999 study period. Figure 2 shows the temporal coverage of the 11 chosen palaeoclimate records.



- *Hydroclimatic significance*. Records used need to provide (i) hydroclimatic information such as precipitation or streamflow or (ii) information that is related to hydroclimatic variability in eastern Australia (e.g. reconstructions of influential large-scale ocean-atmospheric processes such as the El Niño/Southern Oscillation (ENSO), the
Interdecadal Pacific Oscillation (IPO), or the Pacific Decadal Oscillation (PDO)).
- *Availability and accessibility.* Records used need to be readily available (i.e. published) as dated time series so that wet/dry epochs can be easily determined.

**Table 1: Published palaeoclimate records used in this study.**

| | Source of record | Relevant location within eastern Australia | Authors | Temporal resolution | Further details |
|---|---|---|---|---|---|
| Local records | Speleothems (cave deposits) | Wombeyan Caves near Sydney, NSW | Ho et al. (2015b), based on earlier work by McDonald et al. (2007) | Annual | We use the annually resolved two-state aridity index (dry or not dry) |
| | Tree rings | Lamington National Park in southeast Queensland | Heinrich et al. (2009) | Annual | Reconstructed precipitation (wet or dry) |
| | Tree rings | Western Tasmania | Allen et al. (2015a,b) | Annual | Reconstructed streamflow (wet or dry) |
| | Lake sediments | Lake Elingamite, northern Victoria | Barr et al. (2014) | ~4 years | Diatom (microfossil) presence and abundance used to infer low (wet) or high (dry) conductivity |
| | Lake sediments | Lake Surprise, northern Victoria | Barr et al. (2014) | ~5 years | |
| Remote records | Ice cores | Eastern Australia | Vance et al. (2015) | Annual | Reconstructed IPO and associated rainfall variability |
| | Corals | Tropical eastern Australia | Hendy et al. (2003); Lough et al. (2011) | Annual | Presence (wet) and absence (dry) of luminescence lines; Reconstructed rainfall anomalies (positive=wet, negative=dry) |
| | Multiproxy | Murray-Darling River | Gallant and Gergis (2011) | Annual | Reconstructed streamflow |
| | Multiproxy | Murray-Darling River | McGowan et al. (2009) | Annual | Reconstructed streamflow |
| | ENSO/PDO reconstruction | Eastern Australia | Verdon and Franks (2006) | Annual | Reconstructed ENSO/PDO and associated rainfall variability |
| | ENSO/IPO reconstruction | Eastern Australia | Buckley et al. (2019) | Annual | IPO reconstructed from trans-Pacific tree rings |







**Figure 1: Location map of selected palaeoclimate reconstructions for Queensland (QLD), New South Wales (NSW), the Australian Capital Territory (ACT), Victoria (VIC), Tasmania (TAS) and South Australia (SA). For remote palaeoclimate records the map indicates where the studies listed have found significant relationships/impacts while for local records the map shows the actual location of the proxy (refer to Table 1 for further details on the local and remote palaeoclimate records used in this study).**





| | Source of record | Relevant location within eastern Australia | 1000s | 1100s | 1200s | 1300s | 1400s | 1500s | 1600s | 1700s | 1800s | 1900s |
|---|---|---|---|---|---|---|---|---|---|---|---|---|
| Local records | Speleothems (cave deposits) | Wombeyan Caves near Sydney, NSW | ****** | ****** | ****** | ****** | ****** | ****** | ****** | ****** | ****** | ****** |
| | Tree rings | Lamington National Park in southeast Queensland | | | | | | | | | *** | ****** |
| | Tree rings | Western Tasmania | | | | | | ***** | ****** | ****** | ****** | ****** |
| | Lake sediments | Lake Elingamite, northern Victoria | ****** | ****** | ****** | ****** | ****** | ****** | ****** | ****** | ***** | |
| | Lake sediments | Lake Surprise, northern Victoria | ****** | ****** | ****** | ****** | ****** | ****** | ****** | ****** | ****** | ****** |
| Remote records | Ice cores | Eastern Australia | ****** | ****** | ****** | ****** | ****** | ****** | ****** | ****** | ****** | ****** |
| | Corals | Tropical eastern Australia | | | | | | | *** | ****** | ****** | ****** |
| | Multiproxy | Murray-Darling River (Gallant & Gergis, 2011) | | | | | | | | *** | ****** | ****** |
| | Multiproxy | Murray-Darling River (McGowan et al., 2009) | | | | | ***** | ****** | ****** | ****** | ****** | ****** |
| | ENSO/PDO reconstruction | Eastern Australia (Verdon and Franks, 2006) | | | | | | | *** | ****** | ****** | ****** |
| | ENSO/IPO reconstruction | Eastern Australia (Buckley et al., 2019) | | | | *** | ****** | ****** | ****** | ****** | ****** | ****** |

**Figure 2: Time period covered by the published palaeoclimate records used in this study.**

## 3 Creating a composite index of wet/dry epochs based on where/when the majority of pre-instrumental records agree

Each of the selected palaeoclimate records shown in Table 1, Figure 1 and Figure 2 were analysed to determine the occurrence of wet, dry and neutral epochs based on a 5-yearly temporal resolution. The focus is on prolonged (5-yearly or greater) wet and dry periods as these are the most challenging for water resource management and planning (Kiem and Franks, 2004; Johnson et al., 2016; Kiem et al., 2016).

The process for determining wet, dry and neutral 5-year period involved comparing the value for each 5-year period from an individual palaeoclimate record with the average value across the entire period that palaeoclimate record had information available for. If the value for a given 5-year period was more than 20% above (below) average then that 5-year period was classed as wet (dry). All other 5-year periods were classed as neutral (i.e. neither wet or dry) (refer to Section S1 in Supplementary Material for the complete wet/dry/neutral time series for all selected palaeoclimate records).

The insights emerging from each individual palaeoclimate record were then compared to identify if/when there was consensus about hydroclimatic conditions. As documented in previous studies (e.g. Tozer et al., 2016, 2018; Dixon et al., 2017, 2019; Zhang et al., 2018), and as evident from Section S1 in the Supplementary Material, there are numerous sources of uncertainty associated with palaeoclimate records and lack of agreement across different sources of palaeoclimate





information is common. However, there are also instances of agreement and it is these periods where multiple lines of independent evidence agree that we focus on here.

A wet/dry composite index is developed which identifies 5-year periods that the majority of palaeoclimate records analysed here agree were wet or dry (Figure 3). Where there is a lack of information or where there is no clear agreement the 5-year period is classed as Neutral to indicate it is unclear what the hydroclimatic conditions were during that 5-year period (i.e. no consensus across the palaeoclimate records analysed). Three versions of the wet/dry composite index were calculated: one using all palaeoclimate records (top of Figure 3); one using just local palaeoclimate records (middle of Figure 3); and one
using just remote palaeoclimate records (bottom of Figure 3). It is acknowledged that more sophisticated methods for using palaeoclimate data exist and that the composite index of 5-year wet/dry epochs removes the ability to gain insights into (a) interannual variability (e.g. a single very wet year in an overall dry spell or vice-versa) and (b) spatial variability of hydroclimatic conditions across the study area (e.g. a palaeoclimate record correctly identifying wet conditions for a certain location in the study area and another palaeoclimate record correctly identifying dry conditions for a different location
results in an unrealistic classification of neutral conditions across the whole study area). However, while interannual and spatial variability across the study area is important, the focus here is on (a) persistent (i.e. decadal) wet/dry epochs as they are what cause the most problems for water resources management and planning and (b) significant climate events (or shifts) that affect the whole of eastern Australia (i.e. the climate events detected in multiple different types of palaeoclimate records even though the spatial extents (or relevant locations) for the palaeoclimate records are very different (as illustrated in Figure
1)).

To verify the reliability of each wet/dry composite index, we assessed how closely they represent conditions in the instrumental period. The majority of palaeoclimate records clearly identify the second half of the Federation drought (1895-1902), the extremely dry periods of the late 1960s and 1982-1983 (Verdon-Kidd and Kiem, 2009), and the dry IPO-positive
phase between ~1915 and ~1942. Major wet periods can also be seen, such as the 1950s and 1970s during which several major flood events occurred across New South Wales and Queensland (especially during the La Niña years of 1956 and 1974) (e.g. McMahon and Kiem, 2018; Holland et al., 1987). Given the palaeoclimate data (and the resulting wet/dry composite index) satisfactorily identify known instrumental wet and dry periods, it is reasonable to assume that the pre-instrumental parts of the wet/dry composite index are also realistic. Note however that this is based on the assumption of
stationarity, which is a potentially flawed but currently necessary assumption required when reconstructing pre-instrumental data from palaeoclimate proxies (Gallant et al., 2013).

Also highlighted by Figure 3 is that local and remote reconstructions do not always give the same result. This is expected since remote proxies give information about overarching conditions for a large area (e.g. an ENSO reconstruction) and
frequently explain less of the variability in the region of interest than local proxies which record specific information for the



local region where the proxy is sourced from (or directly applicable to). This emphasizes that, whenever possible, multiple proxies from both remote and local sources should be considered and that care should be taken when looking to infer location-specific information from remote proxies, or any aggregation that includes local or remote proxies that are not specifically developed for the target location. Where and when remote and local proxies disagree also raises interesting

questions about why this is the case and points to areas where improvement in our understanding of climate dynamics is required (e.g. Gallant et al., 2013) – this is further explored in Section 4.3.2.

Despite some obvious differences between the three versions of the wet/dry composite index there is also some coherence. All three versions of the wet/dry composite index show a generally dry period from ~1000-1150 and a wet period from

~1550-1600. Both remote and local proxies also indicate that there have been wet and dry epochs in the past that have persisted for significantly longer than wet and dry epochs in the instrumental period (1900-1999). Therefore, despite some differences between the remote and local proxies, the results suggest that the instrumental period may not be a true indicator of the potential for prolonged wet and dry conditions in eastern Australia. Another interesting feature, demonstrated in all three versions of the wet/dry composite index, is that the past ~200 years appear to show the most variability (i.e. the most

frequent shifts between wet and dry conditions). This is consistent with the findings from other studies (e.g. Zhang et al., 2018; Dixon et al., 2019) and again suggests that the last ~200 years is unusual and that multidecadal-scale persistence rather than more frequent fluctuations between wet and dry conditions has dominated most of the last 1000 years.



**Figure 3: Wet/dry composite index for all palaeoclimate records (top), just local records (middle), just remote records (bottom).**
**Number of records available for each 5-year period is also shown (11 records in total were used in this study as per Table 1).**





## 4 Results

### 4.1 Frequency of pre-instrumental wet and dry epochs

The frequency of wet and dry epochs was determined by dividing the total wet/dry composite time series (top of Figure 3) into ten 105-year periods (starting every 100 years from 1000 to 1900). Overlapping 105–year periods were chosen in order

to identify any wet or dry events that may have occurred across the turn of the century, such as the Federation and Millennium droughts. This allowed for a calculation of the relative difference and frequencies of wet and dry epochs of the pre-instrumental and the most recent 105-year instrumental period (1895-1999).

Figure 4 shows the percentage of dry, neutral and wet years for each 105-year period between 1000 and 1999. It is clear that

the most recent 105-year period (1895-1999) does not accurately reflect conditions in any of the other centuries, and that none of the 105-year periods are particularly alike (with the possible exception of 1300-1405 and 1400-1505). The 1000s, 1100s and 1400s all experienced a much greater proportion of dry years than the instrumental period, as well as no wet years, while the 1500s, 1600s, 1700s and 1800s all recorded a greater proportion of wet years than the instrumental period. Water resource managers therefore need to be cautious when using the instrumental period as a basis for planning and

infrastructure design since it is clear that the 105 years from 1895 have a very different distribution of wet and dry years to the previous nine centuries. Similar conclusions and recommendations were also made in Tozer et al. (2016) for a different study location and using a different methodology and approach.

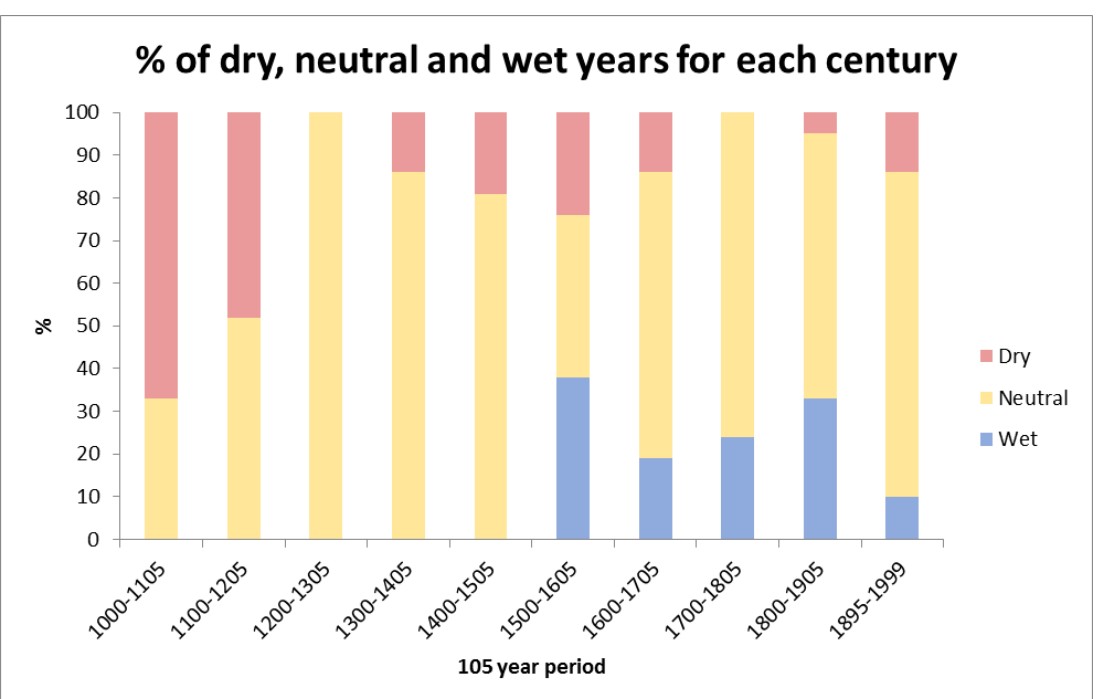

**Figure 4: Proportion (%) of dry, neutral and wet years for each 105-year period starting every 100 years from 1000 to 1895.**



## 4.2 Duration of pre-instrumental wet and dry epochs

For all individual palaeoclimate records, the instrumental period was compared to the pre-instrumental period to determine if any pre-instrumental conditions have an instrumental period analogue. Instrumental period analogues were identified by listing, for each 5-year instrumental period (i.e. 1900-1904, 1905-1909, …), the 5-year pre-instrumental periods in which all palaeoclimate records had exactly the same wet, dry or neutral classification (as per Section S1 in Supplementary Material). Figure 5 shows, where possible, the 5-year instrumental periods that were determined to be equivalent to the pre-instrumental 5-year periods (Section S2 in Supplementary Material lists the actual periods). Figure 5 shows that every 5-year instrumental period had at least one equivalent in the pre-instrumental information. However, while there were 53 5-year pre-instrumental periods identified which did have an instrumental equivalent (29% of the pre-instrumental study period), the majority do not have any instrumental analogue. In other words, 71% of the pre-instrumental period was found to have no equivalent instrumental periods. Hence, for most 5-year pre-instrumental periods there is no modern equivalent captured by the instrumental record, meaning that the vast majority of the 900 years from 1000-1899 was unlike anything that has been encountered in the period on which all water infrastructure, planning and policy is based.

Where there were a number of consecutively similar periods, duration was calculated, allowing for an assessment of whether conditions experienced in the instrumental period have persisted for longer during the pre-instrumental period. Figure 6 shows the duration of pre-instrumental periods found to be equivalent to certain instrumental 5-year periods. As Figure 6 demonstrates, similar pre-instrumental periods have durations ranging from 5 to 70 years.





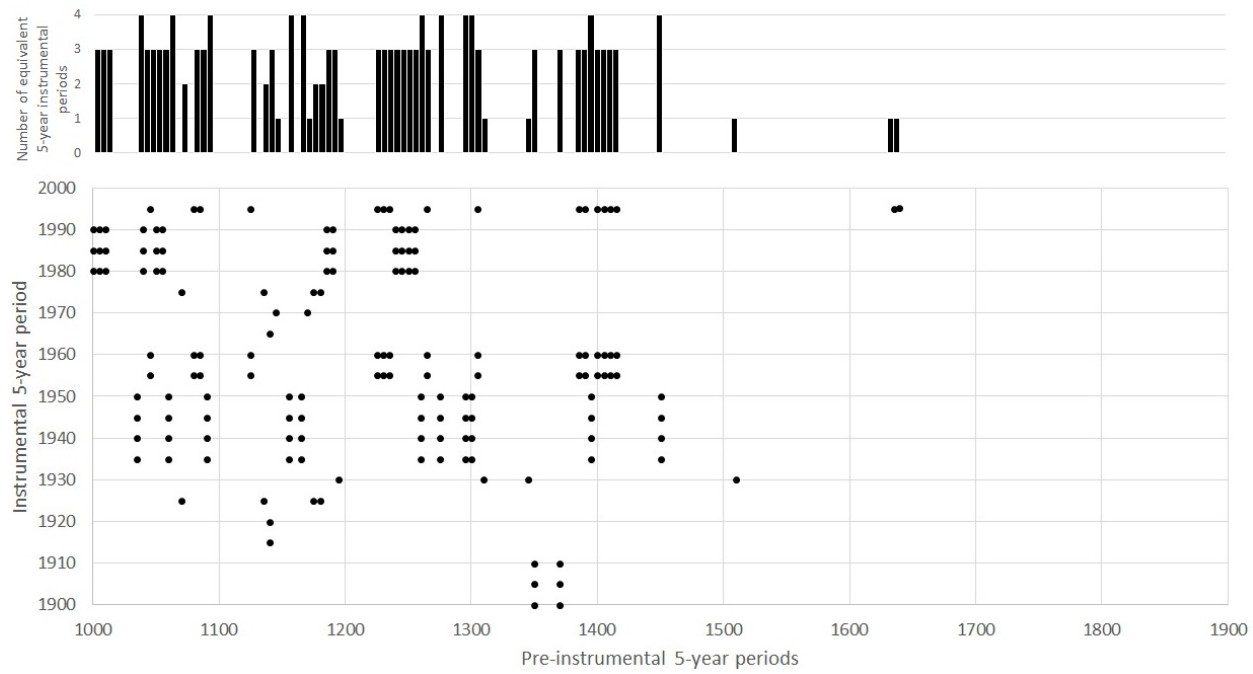

**Figure 5: Instrumental equivalents to pre-instrumental 5-year periods.**

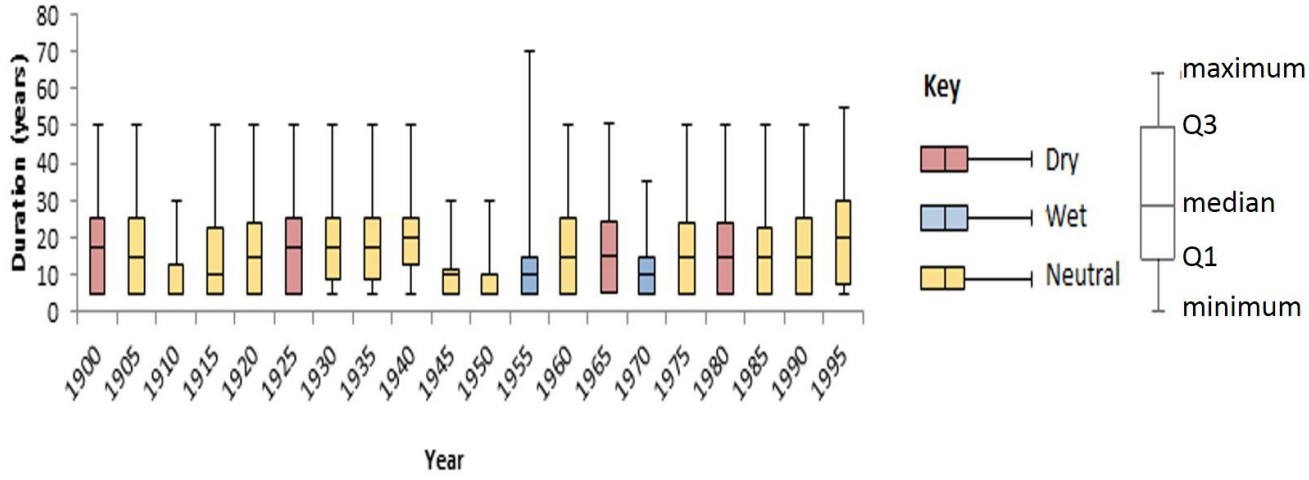


**Figure 6: Duration of pre-instrumental periods with similar hydroclimatic conditions as 5-year periods from the instrumental record. Instrumental periods identified as being wet, dry or neutral are coloured correspondingly.**





The 1900-1904, 1925-1929 and 1980-1984 periods were all considerably dry. As Figure 6 shows, pre-instrumental conditions similar to these periods have lasted for up to 50 years in the past. For example, 16 periods greater than 5 years

between 1000 and 1899 were found that are similar to the dry 1900-1904 period (a period associated with the well-known 1895-1902 Federation drought). This suggests that dry conditions similar to those experienced during the Federation drought have occurred previously for up to 10 consecutive 5-year periods, and that there is potential for dry periods of similar duration to occur again. Due to the variables reconstructed in some of the proxy studies, the magnitude of these dry periods cannot be determined using these data. However, even slightly drier than average conditions for these durations can have

significant effects on water supplies, the environment and socioeconomic conditions (e.g. Kiem, 2013; Kiem and Austin, 2013; Kiem et al., 2016).

Similarly, 1955-1959 is a known wet period that was associated with significant flooding across much of eastern Australia. Fifteen periods in the palaeo-record were identified between 1000 and 1899, which demonstrated a similar combination of

records as 1955-1959 but had durations ranging from 1 to 14 5-year periods (i.e. 5 to 70 years). Given wet epochs of these durations have occurred in the past, there is potential for similar to happen again in the future. Again, while it is not possible to determine the upper magnitude of these wet events, wet conditions that persist for decades can lead to serious water management issues and significantly elevated flood risk due to the role antecedent catchment conditions play in modulating flood risk (e.g. Kiem et al., 2006; Pui et al., 2011; Johnson et al., 2016).

**4.3    Spatial extent of pre-instrumental wet and dry epochs**

Australian Water Availability Project (AWAP) (Jones et al., 2009) rainfall data was used to produce rainfall anomaly maps for eastern Australia. AWAP data is an Australia wide gridded rainfall dataset available as monthly rainfall averages and anomalies, extrapolated from gauged daily or monthly rainfall data (Tozer et al., 2012; King et al., 2013). Annual gridded AWAP data was obtained and combined into 5-year periods from 1900 to 1999 (1900-1904, 1905-1909, …) to produce

maps showing 5-yearly totals relative to the total 5-yearly average over the instrumental period when AWAP data is available (refer to Section S3 in Supplementary Material for the maps for each 5-year period).

The pre-instrumental 5-year periods previously identified as being equivalent to specific 5-year instrumental periods were then matched to the corresponding maps created using AWAP data. The spatial extent of individual palaeoclimate

reconstructions indicating either wet or dry conditions was overlaid on the AWAP maps to give an indication of the spatial extent of pre-instrumental wet and dry epochs. Where the spatial extent of palaeoclimate reconstructions matched actual rainfall patterns for 5-year instrumental periods, there is a chance that the pre-instrumental hydroclimatic conditions were also similar. One wet and one dry period will be examined in closer detail.





### 4.3.1    Pre-instrumental dry epochs similar to the 1965-1969 drought experienced across most of eastern Australia

The late 1960s were associated with serious drought conditions for much of eastern Australia. As shown in Figure 7a, this drought was very widespread but particularly affected eastern Australia between 1964 and 1968. The 1965-1969 drought is chosen rather than the more iconic Federation, World War II or Millennium droughts (Verdon-Kidd and Kiem, 2009) because (i) instrumental data records are incomplete for the Federation and World War II droughts and (ii) the majority of palaeoclimate records used in this study do not cover the full duration of the Millennium drought.


Our results suggest that the period from 1140 to 1144 is equivalent to 1965-1969. Palaeoclimate records for both of these periods indicate drier conditions, particularly the Wombeyan record (Ho et al., 2015b), the ice core record (Vance et al., 2015) and the Barr et al. (2014) lake sediment records. The areas covered by these records, the NSW southern tablelands, southeast Queensland and the Murray River respectively, can be seen on Figure 7a as some of the driest areas recorded

between 1965 and 1969. The number of similarities found between records when comparing 1965-1969 and 1140-1144 suggest that 'extreme' dry or wet conditions encountered in the instrumental period are not unprecedented. Additionally, this shows that it is possible to estimate the spatial extent of pre-instrumental wet and dry periods if there is a modern analogue available to base an estimate on.

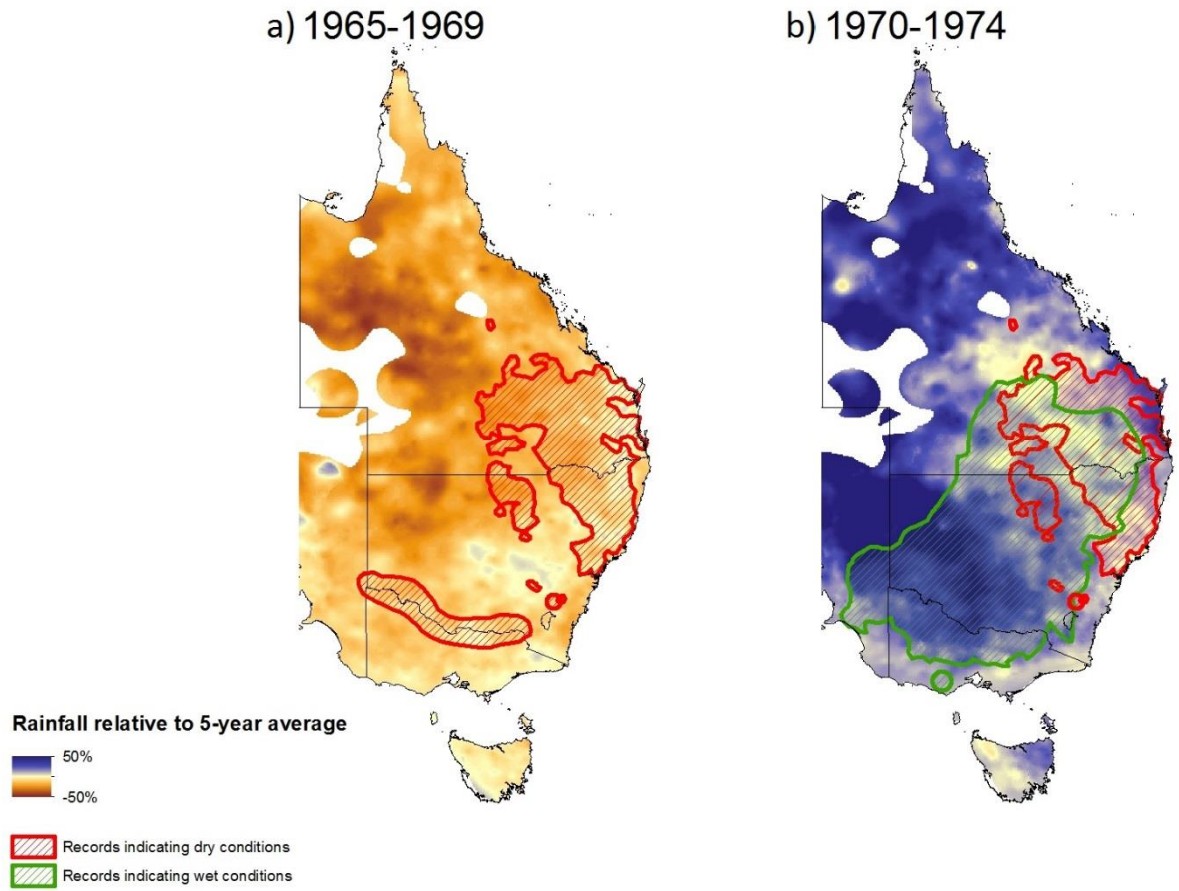

**Figure 7: 1965-1969 and 1970-1974 AWAP maps, showing spatial extent of wet/dry conditions suggested by palaeoclimate reconstructions of corresponding pre-instrumental periods.**

### 4.3.2 Pre-instrumental wet epochs similar to the flood-dominated 1970-1974 period experienced across most of eastern Australia

The period between 1970 and 1974 was a strong La Niña period with above average rainfall for most of eastern Australia. A number of tropical cyclones combined with excessive rainfall caused flooding in Brisbane and Tasmania, and Victoria also experienced its highest recorded rainfall. Figure 7b clearly shows widespread, above average rainfall during this 5-year period.

The majority of local, remote and total records for 1970-1974 all indicate wet conditions. According to previous analysis, 1145-1149 and 1170-1174 also experienced similar rainfall conditions. However, the majority of records for these periods do not agree on persistent wet conditions. In 1170-1174, Ho et al. (2015b) and Vance et al. (2015) suggest conditions were dry. However, Figure 7b shows that these records do in fact cover the regions of eastern Australia that were drier between 1970





and 1974. Similarly, in 1170-1174 Barr et al. (2014) (Lake Surprise) show wet conditions, and these overlap with the wetter regions in Figure 7b. Similar trends can also be seen with the other equivalent period, 1145-1149. A low resolution eastern
Australian record (not used here because it did not meet the temporal resolution criteria) from Stradbroke Island in eastern subtropical Australia also suggests the mid-1100s were particularly dry (Barr et al., 2019).

This further validates the methods used in this study, and shows that the wet/dry composite index displayed in Figure 3 is likely a conservative estimate of the actual pre-instrumental wet and dry conditions. If this is indeed the case, there is a high
chance that 'extreme' rainfall and flood conditions experienced between 1970 and 1974 may have been surpassed by wetter events in the past. As discussed previously, only 29% of the pre-instrumental study period was found to have an instrumental equivalent, meaning that conditions for the majority of 1000-1899 were unlike anything experienced in the instrumental period. At this stage there is no way of determining the spatial signature of events that do not have an instrumental analogue, making it difficult to estimate exactly which areas would be impacted if/when those pre-instrumental wet or dry conditions
return. What is certain, however, is that existing water resource planning, infrastructure and management has been developed based on only 29% of the hydroclimatic variability that is possible. Furthermore, the reliability of existing water resource and water hazard management systems under conditions experienced in the remaining 71% of the last 1000 years is currently unknown – and this represents a significant source of vulnerability to our environmental and socioeconomic sustainability.

## 5   Implications for water resource management and planning: A case study using a typical water storage reservoir
from eastern Australia

The preceding results showed that there is potential for more extreme and prolonged wet and dry epochs to occur throughout eastern Australia. To gain more of an understanding into exactly what impact these events may have on catchment-scale water resources, the 1000-year "total records" wet-dry composite index (Figure 3, top) is used in a model that estimates the water stored (as a percentage of total capacity) in a water storage reservoir. The water storage reservoir is typical of most
dams in eastern Australia and it, and the water storage model, is further described in Kiem and Franks (2004). Annual dam capacity is influenced by inputs (rainfall, catchment runoff and inflow via pumping stations) and outputs (evaporation, spill and supply to the population). This analysis has assumed conservative conditions with an initial capacity of 100%. According to Australian standards, dam levels must not be under the critical threshold (30%) for more than 1% of the time (Kiem and Franks, 2004).


Based on annual rainfall, the averages for the wettest, driest and middle input and output values were calculated to give an indicative value for a typical 'wet', 'dry' and 'neutral' period. These values are then applied to the 1000-year wet-dry composite index previously developed to produce a 1000-year reconstruction of the case study's water storage levels (see Figure 8). While this dam obviously has not existed for the past 1000 years, this reconstruction provides insight into how





water security in the region could be affected if some of the pre-instrumental conditions described previously were to occur
again.

Figure 8 clearly identifies the Federation drought as a time of potentially lower dam levels as well as an intense drought
experienced by eastern Australia in 1982-1983 (Verdon-Kidd and Kiem, 2009). This highlights the model's ability to portray

realistic results throughout the instrumental period, and the limited range of drought and flooding experienced since 1900,
relative to the pre-1900 conditions. This is highlighted by two examples:

1.  The first 200 years (1000-1200) were significantly drier than the most recent century and during this period dam
level was constantly below the critical threshold of 30% capacity.

2.  Apart from dry epochs in the late 1300s, mid-1400s to the mid-1500s, 1620-1650 and late 1600s, most of the period

post-1200 was associated with wet conditions, with the case study showing several extended periods at capacity
(e.g. almost 200 years between ~1700 and 1900 where the reservoir would have been spilling).

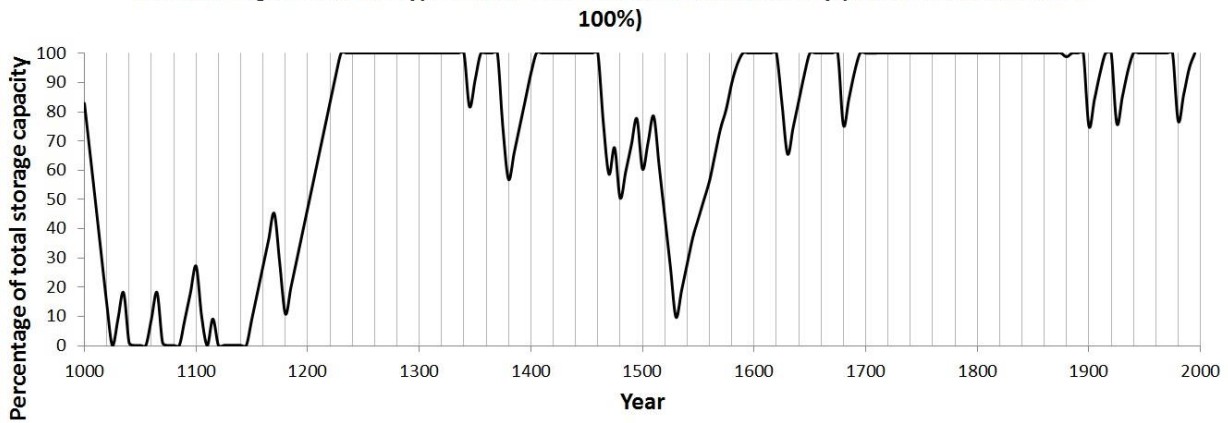

**Figure 8: Case study reservoir water storage levels based on initial storage in CE 1000 equal to 100% of capacity.**

Figure 8 shows that this case study reservoir was under the critical threshold of 30% capacity for 18.5% of the past 1000
years. Therefore, if conditions similar to any one of these centuries were to take place again, this water catchment area would
be placed under significant stress and would need to implement additional water conservation and supply measures.

On the other hand, a dam at capacity for an extended period of time could also pose major issues. For example, Wivenhoe

Dam in southeast Queensland reached capacity during the 2010-2011 Queensland floods and its controlled water release (i.e.
spilling) was associated with extensive flooding in the Brisbane area, destroying many homes and businesses
(http://www.floodcommission.qld.gov.au/) (Van den Honert and McAneney, 2011; McMahon and Kiem, 2018). According
to Figure 8 there were many times in the past 1000 years where the case study reservoir used here would have been at





capacity, and frequent spilling, often for extended periods, would have been necessary. Whether or not this would lead to
flooding and damages similar to, or worse than, that seen in southeast Queensland in 2010-2011 is a question that requires
further investigation. Nevertheless, the durations the reservoir has been at capacity pre-1900 far exceed anything observed in
the instrumental period and therefore such situations are unlikely to be accounted for in existing design, planning and
management.

When combined with the projected increases in population and demand for water, it is likely that droughts and floods will
have an even greater impact on eastern Australia's water supplies. This again highlights how underprepared water
management authorities are with regard to extreme and prolonged wet/dry periods like those identified in the palaeoclimate
record.

Though our case study location is eastern Australia, these challenges, and the limitations associated with current methods
that depend on instrumental records that are too short to realistically characterise interannual to multidecadal variability, also
apply globally. For example, as summarised in the Introduction, palaeoclimate reconstructions from many places around the
world (e.g. non-eastern Australia, Asia, USA, Europe, South America) also demonstrate that, as with eastern Australia, the
relatively short instrumental hydroclimatic records (typically available from ~1900 at best) misrepresent the range of
hydroclimatic variability that is possible and the risks associated with extreme hydrological events (e.g. prolonged drought or
flood dominated epochs). The inadequacies of relying on relatively short instrumental record for drought risk quantification
and management and for flood frequency analysis is increasingly being recognised by practitioners. For example, in 2019 the
USA Geological Survey released new guidelines for flood frequency analysis (England Jr et al., 2019), the first update in 37
years, that documented for the first time how historical and palaeoclimate evidence should be used in flood frequency
analysis and flood risk assessments. Canada (Natural Resources Canada, 2019) and Australia (in the recently updated
Australian Rainfall and Runoff, http://arr.ga.gov.au/) also recognise the need to better account for interannual to
multidecadal variability beyond that seen in the instrumental records but limited details are given on how to do this.

## 6   Conclusions

This study suggests that only 29% of the pre-instrumental period is equivalent to conditions experienced in the instrumental
period. This means that 71% of the pre-instrumental study period has no instrumental equivalent with which to compare it.
Therefore, the most important, and concerning, finding from this study is that the range of hydroclimatic conditions
experienced in the instrumental period is not indicative of the broader 1000-year period. The proportion, frequency and
duration of wet and dry events in the pre-instrumental period is mostly unlike anything experienced instrumentally and the
pre-instrumental records also identify much more severe and prolonged wet/dry epochs. Given that current water resource
management strategies are based on the instrumental period only, eastern Australia is probably not equipped to manage





water resources during prolonged (i.e. decadal-scale) wet/dry epochs that have occurred in the pre-instrumental period. When coupled with projected impacts of climate change and the demands of a growing population, the impacts of future 'extreme' wet or dry events will likely be significantly greater and more widespread. This represents a significant challenge for water resources and water hazard management in Australia and highlights that infrastructure design and adaptation

strategies are probably not as resilient or secure as is suggested by instrumental-record based risk assessments.

Though our case study location is eastern Australia, there is also evidence in the literature that short instrumental hydroclimatic records misrepresent the range of hydroclimatic variability and risk that is possible in, for example, non-eastern Australia, Asia, USA, Europe, and South America. These realities require a paradigm shift from current practice that

assumes probability models calibrated to short instrumental records realistically account for the worst dry or wet epochs possible and that the chance of drought or flood risk does not change over time. These assumptions are clearly incorrect, and leave water supply managers without the tools to properly deal with multidecadal climate variability. Numerous sources of pre-instrumental (palaeoclimate) data (e.g. tree-rings, corals, speleothems) have emerged over the last decade and more are being worked on currently (e.g. refer to Section 4.2 and the Supplementary Material associated with Kiem et al. (2016) for a

detailed review and comparison of existing palaeoclimate information relevant to drought in Australia). Palaeoclimate records can extend hydroclimatic records by centuries or millennia, which, as demonstrated here via a simple case study and a deliberately unsophisticated use of palaeoclimate records, can provide new information about what hydroclimatic conditions are plausible (which is potentially of great value to water resources managers and planners). However, palaeoclimate-based reconstructions of past hydroclimatic variability come with their own uncertainties, assumptions, and

limitations (e.g. Tozer et al., 2016, 2018; Dixon et al., 2017, 2019; Zhang et al., 2018). Consequently, as also recommended in Kiem et al. (2016), further research is needed to (i) better understand, quantify, and deal with uncertainties and inconsistencies in the palaeoclimate information and (ii) provide the translational science required for water resources managers and planners to maximise the practical value of palaeoclimate records in assessing and managing drought and flood risks.

## 7   Code availability

No code was developed or used in this study.

## 8   Data availability

All data used in this paper is publicly available. The palaeoclimate records used were selected from palaeoclimate information available (i.e. published) at the time of study. Refer to Section 2 for details on where to access the palaeoclimate



information used. Australian Water Availability Project (AWAP) rainfall data is available from the Australian Bureau of Meteorology.

## 9 Author contribution

Anna L. Flack: review of palaeoclimate literature to identify palaeoclimate data to use, collection of palaeoclimate data that met the criteria for use in this study, analysis, assist with writing of original draft and subsequent revisions and finalisation.

Anthony S. Kiem: conceptualisation, methodology, analysis, assist with writing of original draft and subsequent revisions and finalisation.

Tessa R. Vance: conceptualisation, methodology, collection/development/provision of ice core information used in the rainfall reconstruction, writing of original draft and subsequent revisions and finalisation.

Carly R. Tozer: methodology, analysis, writing – reviewing and editing.

Jason L. Roberts: conceptualisation, methodology, collection/development/provision of ice core information used in the rainfall reconstruction, writing – reviewing and editing.

## 10 Competing interests

There are no competing interests to declare.

## 11 Acknowledgements

The authors would like to thank Peter Briggs (CSIRO) for supplying the AWAP data. The work conducted to produce this paper was funded by:
- Australian Research Council Discovery Project on "Flooding in Australia – are we properly prepared for how bad it can get?" (ARC DP180102522).
- Australian Research Council Special Research Initiative for Antarctic Gateway Partnership (Project number:
SR140300001).

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
