# Peer review of "Comparison of published palaeoclimate records suitable for reconstructing annual to sub-decadal hydroclimatic variability in eastern Australia: implications for water resource management and planning"

_Hydrology and Earth System Sciences, 2020_

## Referee Comment (RC1) · Anonymous Referee #1 · 27 Jul 2020

This paper provides very interesting results, and topic of the research is within the scope of this journal. The manuscript is well written and organized. Some minor revisions are recommended.

[General Comments]

1. P7, L135 The advantage of using "composite index" is a little unclear. We can understand that the composite index would give conservative estimate of wet/dry condition (as described in P16, L294), however, it is still unclear why composite index

would be better rather than the use of "best-perform single palaeoclimate records". # no palaeoclimte records could be selected as "best-perform"(?)

2. P6, L131 "... uncertainty associated with palaeoclimate records ..." –> It would be better to describe (explain) a little more about uncertainty and/or accuracy (precision) of paleoclimate records, if possible. (which sources of uncertainty and how large/small uncertainties are expected, difference in precision between older records (-1000years) and recent (after 1900), ...) # The response to this comment is not mandatory, however, this information would be helpful # for reader's understanding of characteristics (limitation) of palaeoclimate records.

[Specific Comments]

1. P3, L85 "11 palaeoclimate records that were selected ..." –> How many "candidates" of palaeoclimate records were reviewed (in total) to select 11 palaeoclimate records?

2. P3, L92 "... however, two records with –4-5 year temporal resolution ..." –> How did you use (and composite) 4-5 years temporal resolution data with other annual resolution data? # ex. interpolated in annual resolution (?)

3. P6, L124 "... more than 20% above (below average) ..." –> How to set the "20%" as a threshold? # following to other past research (?)

4. P7, L135 "... majority of palaeoclimate records analyzed here agree were wet or dry ..." –> - "majority" means that if 6 of palaeoclimate records show the signal of wet/dry, the period is considered as wet/dry period. (?)

- Does the number of agreed palaeoclimate records (wet or dry, among 11 records) have any relationship with degree (severity) of wetness/dryness of the period?

5. P16, L316 "Based on annual rainfall ..." –> Which rainfall data was used to calculate averages for wettest, driest and middle input? # AWAP data (?)

314, 2020.
Interactive
comment

---

## Referee Comment (RC2) · Lisa Davis (Referee) · 2 Sep 2020

General Comments (overall quality)

This paper examines decadal and sub-decadal hydroclimatological changes in eastern Australia by performing a metanalysis or synthesis of pre-existing multi-proxy paleo-records from within or in proximity to the region. The results of the paleorecord analyses are applied within the context of a water resources management framework. This paper does several things that make it a novel and timely contribution of broad interest

to many communities (including the paleoenvironmental, hydrologic, hydroclimatologic, and water resources communities) and a good fit for HESS, with its integrative perspective as a journal.

Although the number of regional and continental scale syntheses of paleoenvironmental data have increased over the last decade, too few exist for many locations in the world to make these data accessible and viable for use by hydrologists and water resource professionals. This paper helps ameliorate this issue for a large region of the Australian continent. A second contribution of this work is that it presents a methodology for others to follow to increase the number of regional to continental scale interpretations of paleoenvironmental data for the purposes of water resource management. There is a great need for longer records of hydroclimatological data, particularly when it comes to extremes and droughts because 20th century precipitation and flow records, typically used as the basis for forecasting the occurrence of future extremes, is too short to have a statistically relevant number of extreme event observations to make their predictions of extremes reliable. This problem has been documented worldwide and it could be argued all of humanity is at the precipice of a hydrological crisis given how many major infrastructure designs are based on a 20th century record that no longer applies. Many researchers are producing site specific, paleoenvironmental data, spanning millennia and thus a wide range of hydroclimatological regimes. But they are not analyzing and disseminating the results in a framework that would facilitate the adoption of this information by the hydrologic modeling and water resources community. Thus, the importance of this paper is that it demonstrates a method for interpreting and applying paleoenvironmental data to address water resources and hydrologic assessments of extreme events for others to follow.

This paper being published so soon after the revision of flood frequency guidelines (US Geological Survey Bulletin 17C, released in final form in 2017) makes the paper a very timely publication. These guidelines, designed to inform federal water regulators in the U.S. but used the world over, recommend combining paleodata with instrumented pre-

cipitation and streamflow records to improve the reliability of extreme flood prediction.

Specific Comments

1. Introduction - The emphasis of the introduction should be flipped to make the Australia specific information, currently in the first paragraph, be secondary to the information in the bulleted points about the global issue of short, 20th century records being used as the basis of precip and hydo forecasts. As part of making the broader relevance of the paper more apparent, I suggest expanding the bulleted information between 45-55. I think the point made later about the recommendations of the USGS's newly revised flood frequency guidelines (Bulletin 17C) should be introduced in here as well.

2. Table 1: I would state which multiproxy methods were used so that it is easier for the reader to quickly verify that an annual resolution of data applies. For the Gallant and Gergis (2011), for example, I would change to "Tree Rings & Coral."

3. Table 1: Regarding the remote records, McGowan et al. 2009 is not included in the references. It needs to be added.

4. I found a paper by the same lead author (McGowan) about streamflow reconstructions in the MD River Basin: Geophysical Research Letters (GRL) https://doi.org/10.1029/2008GL037049

If the GRL paper is the same used as a data source for the analyses, I'm not sure I agree that it fits the needs of the analysis. The GRL paper reconstructs streamflow for the Murray River in Australia based on a statistical correlation between the instrumented streamflow record and a reconstruction of the Pacific Decadal Oscillation from paleo records in Canada and China. But no paleorecord was used to validate the association between streamflow in the PDO from anywhere near Australia.

I know there are limited datasets to work with, but this seems too indirectly tied to Australia to be meaningful. The other PDO reconstruction data source used (Buckley

et al. 2019) uses paleorecords from across the Pacific and seems more reasonable to include.

I know the Pacific is large and perhaps by having both PDO reconstructions the goal was to cover all of the Pacific?

This is relevant because of the discussion in Lines 158-171, pgs. 7-8 concerning the accuracy of localized vs. remote reconstructions. The PDO reconstruction for the McGowan paper was built on paleorecords that were the most geographically remote of all the data sources. If this is the same paper.

5. What instrumented data were used in the analyses? Was it precipitation or stream-flow or both and how are the instrumentation data distributed over the study area?

Technical Corrections

1. Line 159, pg. 7: commas needed around "however."
* * *

---

## Author Comment (AC1) · 16 Sep 2020

Response to review comments on the paper "Comparison of published palaeoclimate records suitable for reconstructing annual to sub-decadal hydroclimatic variability in eastern Australia: implications for water resource management and planning" by Flack et al. (Manuscript number: hess-2020-314)

**Reply to comments from Reviewer #1 (Anonymous):**

1-1.  *This paper provides very interesting results, and topic of the research is within the scope of this journal. The manuscript is well written and organized. Some minor revisions are recommended.*

**Author Response:** Thanks for the positive comments. The requested revisions are addressed in the responses below.

1-2.  *The P7, L135 The advantage of using "composite index" is a little unclear. We can understand that the composite index would give conservative estimate of wet/dry condition (as described in P16, L294), however, it is still unclear why composite index would be better rather than the use of "best-perform single palaeoclimate records". #no palaeoclimte records could be selected as "best-perform"(?).*

**Author Response:** It is not possible to identify the "best-performing" palaeoclimate record. Each individual palaeoclimate record has strengths and weaknesses and some are more/less applicable for certain locations. As documented in previous studies (e.g. Tozer et al., 2016, 2018; Dixon et al., 2017, 2019; Zhang et al., 2018), and as evident from Section S1 in the Supplementary Material, there are numerous sources of uncertainty associated with individual palaeoclimate records and lack of agreement across different sources of palaeoclimate information is common. However, there are also instances of agreement and it is these periods where multiple lines of independent evidence agree that we focus on here. Therefore the advantage of using a composite index is that it enables the analysis to concentrate only on the dry/wet epochs that we are confident occurred (i.e. that are evident in the majority of individual palaeoclimate records).

1-3.  *P6, L131 "... uncertainty associated with palaeoclimate records ..." –> It would be better to describe (explain) a little more about uncertainty and/or accuracy (precision) of paleoclimate records, if possible. (which sources of uncertainty and how large/small uncertainties are expected, difference in precision between older records (-1000years) and recent (after 1900), ...) # The response to this comment is not mandatory, however, this information would be helpful # for reader's understanding of characteristics (limitation) of palaeoclimate records.*

**Author Response:** The references provided in this section (e.g. Ho et al., 2015a; Tozer et al., 2016, 2018; Dixon et al., 2017, 2019; Zhang et al., 2018) give comprehensive details about the uncertainties associated with palaeoclimate records. It is not appropriate to repeat that information in this paper.

1-4.  *P3, L85 "11 palaeoclimate records that were selected ..." –> How many "candidates" of palaeoclimate records were reviewed (in total) to select 11 palaeoclimate records?*

**Author Response:** We conducted a literature review using "palaeoclimate" and "Australia" as the search terms. More than 100 papers were returned but sometimes multiple papers were based on the same palaeoclimate record so this does not mean that more than 100 palaeoclimate reconstructions are available. Focusing on palaeoclimate information relevant to eastern Australia only, available at sub-decadal temporal resolution, covering the last 1000 years, and related to hydroclimate (as opposed to temperature) reduced the "candidate" records to 24. The 11 chosen were the ones that were available/accessible and in some cases represented an update of earlier work (e.g. Ho et al. (2015b) is based on earlier work by McDonald et al. (2007) and Vance et al. (2015) is an upgrade of Vance et al. (2013)).

Response to review comments on the paper "Comparison of published palaeoclimate records suitable for reconstructing annual to sub-decadal hydroclimatic variability in eastern Australia: implications for water resource management and planning" by Flack et al. (Manuscript number: hess-2020-314)

*1-5. P3, L92 "... however, two records with –4-5 year temporal resolution ..." –> How did you use (and composite) 4-5 years temporal resolution data with other annual resolution data? # ex. interpolated in annual resolution (?)*

**Author Response:** Yes, every year within the 4-5 year block was given the same value.

*1-6. P6, L124 "... more than 20% above (below average) ..." –> How to set the "20%" as a threshold? # following to other past research (?)*

**Author Response:** The 20% threshold was chosen to be consistent with the rainfall decile approach used by the Australian Bureau of Meteorology to determine whether rainfall is above average, average or below average for a given time period and location (http://www.bom.gov.au/climate/glossary/deciles.shtml). Following this approach rainfall in deciles 1, 2, and 3 (i.e. more than 20% below average) is considered dry and rainfall in deciles 8, 9, and 10 (i.e. more than 20% above average) is considered wet.

*1-7. P7, L135 "... majority of palaeoclimate records analyzed here agree were wet or dry..." –> - "majority" means that if 6 of palaeoclimate records show the signal of wet/dry, the period is considered as wet/dry period. (?)*

**Author Response:** Yes, that is correct. Text has been adjusted to clarify as follows (new text is highlighted):
*A wet/dry composite index is developed which identifies 5-year periods that the majority of palaeoclimate records (i.e. 6 or more) analysed here agree were wet or dry (Figure 3).*

*1-8. Does the number of agreed palaeoclimate records (wet or dry, among 11 records) have any relationship with degree (severity) of wetness/dryness of the period?*

**Author Response:** No, the magnitude of the event (i.e. how dry or wet it was) is not assessed here. If a large number of palaeo records agree that a certain epoch was dry (or wet) then we have increased confidence that it was actually dry (or wet) at the locations represented by those palaeo records. So a large number of palaeo records agreeing that it is dry (or wet), even though they sometimes represent different locations, is more indicative of the spatial extent of the dry (or wet episode) rather than the magnitude of the event.

*1-9. P16, L316 "Based on annual rainfall ..." –> Which rainfall data was used to calculate averages for wettest, driest and middle input? # AWAP data (?)*

**Author Response:** Yes, AWAP data.

---

## Author Comment (AC2) · 16 Sep 2020

Response to review comments on the paper "Comparison of published palaeoclimate records suitable for reconstructing annual to sub-decadal hydroclimatic variability in eastern Australia: implications for water resource management and planning" by Flack et al. (Manuscript number: hess-2020-314)

**Reply to comments from Reviewer #2 (Lisa Davis):**

2-1. *This paper examines decadal and sub-decadal hydroclimatological changes in eastern Australia by performing a metanalysis or synthesis of pre-existing multi-proxy paleorecords from within or in proximity to the region. The results of the paleorecord analyses are applied within the context of a water resources management framework. This paper does several things that make it a novel and timely contribution of broad interest to many communities (including the paleoenvironmental, hydrologic, hydroclimatologic, and water resources communities) and a good fit for HESS, with its integrative perspective as a journal.*

*Although the number of regional and continental scale syntheses of paleoenvironmental data have increased over the last decade, too few exist for many locations in the world to make these data accessible and viable for use by hydrologists and water resource professionals. This paper helps ameliorate this issue for a large region of the Australian continent. A second contribution of this work is that it presents a methodology for others to follow to increase the number of regional to continental scale interpretations of paleoenvironmental data for the purposes of water resource management. There is a great need for longer records of hydroclimatological data, particularly when it comes to extremes and droughts because 20th century precipitation and flow records, typically used as the basis for forecasting the occurrence of future extremes, is too short to have a statistically relevant number of extreme event observations to make their predictions of extremes reliable. This problem has been documented worldwide and it could be argued all of humanity is at the precipice of a hydrological crisis given how many major infrastructure designs are based on a 20th century record that no longer applies. Many researchers are producing site specific, paleoenvironmental data, spanning millennia and thus a wide range of hydroclimatological regimes. But they are not analyzing and disseminating the results in a framework that would facilitate the adoption of this information by the hydrologic modeling and water resources community. Thus, the importance of this paper is that it demonstrates a method for interpreting and applying paleoenvironmental data to address water resources and hydrologic assessments of extreme events for others to follow.*

*This paper being published so soon after the revision of flood frequency guidelines (US Geological Survey Bulletin 17C, released in final form in 2017) makes the paper a very timely publication. These guidelines, designed to inform federal water regulators in the U.S. but used the world over, recommend combining paleodata with instrumented precipitation and streamflow records to improve the reliability of extreme flood prediction.*

**Author Response:** Thanks for these positive comments. You summarise the intent of our paper well – to demonstrate how insights from palaeoclimate data can be used to improve hydroclimatic risk assessments and water resources management.

Response to review comments on the paper "Comparison of published palaeoclimate records suitable for reconstructing annual to sub-decadal hydroclimatic variability in eastern Australia: implications for water resource management and planning" by Flack et al. (Manuscript number: hess-2020-314)

2-2. *Introduction - The emphasis of the introduction should be flipped to make the Australia specific information, currently in the first paragraph, be secondary to the information in the bulleted points about the global issue of short, 20th century records being used as the basis of precip and hydo forecasts. As part of making the broader relevance of the paper more apparent, I suggest expanding the bulleted information between 45-55. I think the point made later about the recommendations of the USGS's newly revised flood frequency guidelines (Bulletin 17C) should be introduced in here as well.*

**Author Response:** Agree. Thanks for this suggestion. The Introduction has been revised as suggested.

2-3. *Table 1: I would state which multiproxy methods were used so that it is easier for the reader to quickly verify that an annual resolution of data applies. For the Gallant and Gergis (2011), for example, I would change to "Tree Rings & Coral."*

**Author Response:** Revised as suggested.

2-4. *Table 1: Regarding the remote records, McGowan et al. 2009 is not included in the references. It needs to be added.*

**Author Response:** Apologies, the details for McGowan et al. (2009) have now been added to the references.

2-5. *I found a paper by the same lead author (McGowan) about streamflow reconstructions in the MD River Basin: Geophysical Research Letters (GRL) https://doi.org/10.1029/2008GL037049. If the GRL paper is the same used as a data source for the analyses, I'm not sure I agree that it fits the needs of the analysis. The GRL paper reconstructs streamflow for the Murray River in Australia based on a statistical correlation between the instrumented streamflow record and a reconstruction of the Pacific Decadal Oscillation from paleo records in Canada and China. But no paleorecord was used to validate the association between streamflow in the PDO from anywhere near Australia. I know there are limited datasets to work with, but this seems too indirectly tied to Australia to be meaningful. The other PDO reconstruction data source used (Buckley et al. 2019) uses paleorecords from across the Pacific and seems more reasonable to include. I know the Pacific is large and perhaps by having both PDO reconstructions the goal was to cover all of the Pacific? This is relevant because of the discussion in Lines 158-171, pgs. 7-8 concerning the accuracy of localized vs. remote reconstructions. The PDO reconstruction for the McGowan paper was built on paleorecords that were the most geographically remote of all the data sources. If this is the same paper.*

**Author Response:** We acknowledge the point made by the reviewer and agree that the McGowan et al. (2009) palaeoclimate information is based on the most geographically remote of all the data sources. However, the important point, and the reason the McGowan et al. (2009) record was included, is that it is based on a published reconstruction of the PDO that is considered satisfactory and the PDO is known to have a significant influence on hydroclimatic variability in eastern Australia. Hence the McGowan et al. (2009) record meets the selection criteria used in this study. We agree that you need to be careful when using remote proxies to infer location-specific hydroclimatic information (hence the discussion and caveats given in Lines 158-171).

Response to review comments on the paper "Comparison of published palaeoclimate records suitable for reconstructing annual to sub-decadal hydroclimatic variability in eastern Australia: implications for water resource management and planning" by Flack et al. (Manuscript number: hess-2020-314)

*2-6.  What instrumented data were used in the analyses? Was it precipitation or streamflow or both and how are the instrumentation data distributed over the study area?*

**Author Response:** Precipitation data from the instrumental period (~1900 to present) was obtained from the Australian Water Availability Project (AWAP) (Jones et al., 2009) and used in Section 4.3. AWAP data is an Australia wide gridded (~5km x 5km resolution) rainfall dataset available as monthly rainfall averages and anomalies. AWAP data is produced by extraploating from gauged daily or monthly rainfall data (Tozer et al., 2012; King et al., 2013).

*2-7.  Line 159, pg. 7: commas needed around "however."*

**Author Response:** Revised as suggested.